# Patients' E-Readiness to use E-Health technologies for oral health

**Arishdeep Kaur Jagde**[1]⊙, **Richa Shrivastava**[2]⊙, **Jocelyne Feine**[1]⊙, **Elham Emami**[1]⊙ *

**1** Faculty of Dentistry, McGill University, Quebec, Canada, **2** Faculty of Dentistry, Université de Montréal, Quebec, Canada

⊙ These authors contributed equally to this work.
* elham.emami@mcgill.ca

## Abstract

### Introduction

Scientific evidence highlights the importance of E-Readiness in the adoption and implementation of E-Oral Health technologies. However, to our knowledge, there is no study investigating the perspective of patients in this regard. Therefore, the objective of this study was to explore patients' E-Readiness in the field of dentistry.

### Materials and methods

A qualitative study was conducted using interpretive descriptive methodology. Purposeful sampling with maximum variation and snowball techniques were used to recruit the study participants via McGill University dental clinics and affiliated hospitals, as well as private or public dental care organizations. A total of 15 face-to-face, semi-structured and 60 to 90-minute audio recorded interviews were conducted. Data collection and analyses were performed concurrently, and interviews were continued until saturation was reached. Activity theory was used as the conceptual framework, and thematic analysis was used to analyze data. Data analysis was conducted both manually and with the use of "ATLAS-ti" software.

### Results

Four major themes emerged from the study; unlocking barriers, E-Oral Health awareness, inquisitiveness for E-Oral Health technology and enduring oral health benefits. These themes correspond with all three types of readiness (core, engagement and structural).

### Conclusion

The study results suggest that dental patients consider E-Oral Health as a facilitator to access to care, and they are ready to learn and use E-Oral Health technology. There is a need to implement and support E-Oral Health technologies to improve patient care.

**Data Availability Statement:** All relevant data are within the manuscript and its Supporting Information files.

**Funding:** The authors received no specific funding for this work.

**Competing interests:** The authors have declared that no competing interests exist.

## Introduction

Oral health has been recognized as a fundamental human right, yet more than 50% of the world's population is in need of suitable and affordable oral health care [1]. People with low-incomes, senior citizens, individuals with special needs, new immigrants, refugees, Indigenous peoples and those living in rural and remote areas face disparities and challenges in access to oral health care [2, 3]. Factors such as shortage of oral health care providers and facilities, geographic barriers to access oral health care services and associated costs result in poor oral health [4, 5]. Poor oral health can also be related to cultural and linguistic barriers, poor education and oral health illiteracy [6–8].

The use of E-Health technology has been recognized as an innovative approach to address the challenges in health care systems [9]. E-Health innovation has been defined by Eysenbach *et al.* as '*an emerging field in the intersection of medical informatics, public health and business, referring to health services and information delivered or enhanced through the internet and related technologies'* [10].

E-Health technologies, such as online communities, electronic health records, web portals and telehealth applications, have been used in various disciplines, including dentistry, for disease diagnosis and screening, reducing health illiteracy, optimizing education, facilitating exchange of information and improving communication between patients and health care providers, as well as increasing access to health services [11, 12].

Despite the substantial potential impact of E-innovations on health care, implementation of this technology still faces barriers that include E-Health illiteracy, lack of awareness and readiness, unwillingness to use technology, high cost, need for training and long-term sustainability of E-Health platforms [13–15].

The scientific literature highlights the importance of E-Readiness in the adoption and implementation of E-Health technologies [15]. E-Readiness has been defined as "*the degree to which users, healthcare institutions, and the healthcare system itself, are prepared to participate and succeed with e-health implementation.*" [13]. Jennett *et al.* have introduced three E-Readiness domains: Core readiness, Engagement readiness and Structural readiness. Core readiness refers to "*the need for telehealth services, a dissatisfaction with the status quo and an expectation of change*". Engagement readiness refers to "*understanding as well as assessing the advantages and disadvantages of telehealth*" and structural readiness is "*the development of infrastructure such as adequate human resources, technical structures as well as necessary training for telehealth implementation*" [13].

Accordingly, analysis of the E-Readiness framework revealed that there is a need to develop specific assessment tools for various sectors such as stakeholders, managers, health organizations and health care providers [16]. However, the area of E-Health readiness assessment needs further research before attempts are made to develop a more generic framework for different disciplines [16]. Based on our research, there is no study that specifically examines e-readiness in the discipline of dentistry from a patient's perspective. Therefore, the objective of this study was to explore the readiness of patients to use E-Oral Health care and services.

## Materials and methods

Ethics approval for this study was obtained from the Institutional Review Board of McGill University's Faculty of Medicine (Ethical approval registration number: A11-B63-18B). Signed consent forms were obtained from all study participants. This study used a qualitative and "interpretive description" approach to gain a deep insight into the perceptions of individuals concerning E-Health technology [17]. "Interpretive description" methodology, introduced by Thorne (1997), is suitable for small-scale qualitative studies and for research in the domain of

clinical practice generating clinical practice-based knowledge [17]. It goes beyond the theoretical description of the phenomenon and offers more practical forms of the interpretation [17].

## Study setting, participants, and data collection

The study participants were recruited from dental clinics and affiliated hospitals at McGill University, as well as other private or public health care clinics. The participants were seeking oral health care for themselves, their children or other family members. All participants or their family members from various cultural, educational and socio-economic backgrounds were eligible to be included in the study. A semi-structured interview guide was designed based on the study framework. A purposeful sampling with maximum variation, as well as a snowball sampling technique were used to recruit the study participants [18]. This approach allowed us to collect "information-rich" data and capture the perspectives of a wide range of people, regardless of their backgrounds [18]. By using snowball sampling, the recruited participants were asked to identify other participants who might be interested in participating in the study. Data were collected using in-depth, face-to-face, audio-recorded and 60 to 90-minute interviews. The inclusion criteria to participate in study were age above 18 years, Montreal resident and ability to speak and understand English. The exclusion criteria were the non-willingness of the participant to provide consent. These interviews were conducted by a postgraduate student (AKJ) trained in qualitative research and at a place suited to the interviewee. Data collection and analysis were performed concurrently, and interviews were continued until saturation was reached [19].

## Data analysis

Analysis included transcription, debriefing, codification, data display, inductive-deductive thematic analysis and interpretation [20, 21]. Data were coded manually, then analyzed using "ATLAS.ti" version 8 to facilitate the analysis. The first coding round used the principles of text interpretation developed by Strauss and Corbin (1998). This method involved cutting the transcript into significant sections [22]. We used an initial list of codes inspired by the type of E-Readiness and, throughout the coding, we refined the list. Then, the codes and their respective texts were examined and grouped into broad themes (Table 1. Development of Categories). The preliminary interpretations were reviewed during research team meetings, and themes were elaborated collectively.

## Conceptual framework

The Activity Theory framework adopted for E-Health readiness assessment was used as the conceptual framework for this study, as shown in Fig 1 [23, 24]. The Activity Theory offers a philosophical structure for studying the developmental processes that interlink individuals and society [25]. This sophisticated tool has potential to provide a rich, systematic and more structured description of human activities in any complex and dynamic environment [26].

It provides a helpful paradigm to understanding the meaning of technology for people, including human experience, needs, environment, motivations, complexities and efficiency of emerging technologies [25]. As shown in Table 2, this framework was used in the development of the interview guide, in understanding user behavior and associated broader contextual problems on E-Health technology usability and in analysing the data [23].

## Results

The profile of the study participants is shown in Table 3. Data saturation was reached after the 10th interview; however, data collection continued up to the 15th interview to ensure the

**Table 1. Development of categories.**

| CORE CATEGORY | THEMES | OPEN CODES | QUOTATION |
|---|---|---|---|
| **Core readiness**<br><br>*It assesses the degree to which members of a community are unhappy with their current health care provision, see E-Health as a solution and communicate their need and readiness for E-Health services.* | Unlocking barriers | Participants' dissatisfaction with current health care system | *One or two times I was trying to get an appointment for to get a checkup, because I used to live in Vancouver but in that time, it was like the access wasn't easy, because I have to wait for a long time, maybe six or seven months to get the appointment, then I decided to go back my home country and do it there. So, then I got it done from there.* |
| | | | *I'm really running short of the information where to go Where not to go collect the data where what and when it should be done* |
| | | | *Biggest problem I will tell you, I guess 2 years back, for my kid is having a toothache and it's you know weekend mean to say Friday night it gonna be hard for me to go to a dentist, you know so I have to wait for Monday.* |
| | | Oral health needs | *I expect high standard of high quality of all health care and I think I would say with the most modern life technology, in terms of the quality of healthcare will improve.* |
| | | | *You know, if the dental services are available for all the time like 24* 7, Everybody would love to have those services because the health is the kind of thing, things happens. You don't know the time* |
| | | | *Well, if you can get faster service faster care less complicated, that would be great, save a lot of time.* |
| | | | *I think people should get appointment easily and it should be convenient for the people who are new to Canada and it should be cost effective. So, any person can go for the dental treatment.* |
| | Enduring oral health benefits | Preparedness and E-Oral Health as a solution | *I think to have more access to the E- appraisal of healthcare or Cybernet will be really it will be too good stead for benefits to the society* |
| | | | *In terms of efficiency, there's definitely an improvement that can be done with e health* |
| | | | *it's very advanced, and you can take advantage of using this type of application, it will help a lot* |
| | | | *this is something new and something like do you can say improvement so this is a good idea having you can access your oral health on net.* |
| | | | *it will be helpful for old age people too and international people also, but we cannot implement it hundred percent right now. In future this is going to be the best thing.* |
| **Engagement readiness**<br><br>*It assesses the degree to which a community member is exposed to the concept of E-Health and actively discusses its potential benefits and negative effects. It also includes assessing the ability and willingness of members of a community to accept E-Health training.* | Affordability<br><br>Inquisitiveness of E-Oral Health technology | Understand E-Oral Health advantages and disadvantages | *it is same thing like a taxi, you know where you reach for taxi and do you have the number you have on the Cybernet where to reach for taxi or the or for your breakfast, first it will be same way good and I think it's additional advantage.* |
| | | Willingness to be trained | *you know, as a mother if you ask any mother to this (E-Oral Health) she will say yes, because that's the kind of very handy so I can easily access, I can talk to maybe I can text them this is a problem and what should be the next step* |

*(Continued)*

**Table 1.** (Continued)

| CORE CATEGORY | THEMES | OPEN CODES | QUOTATION |
|---|---|---|---|
| **Structural readiness** *This measures the accessibility and cost of Information and Communication Technology resources that are necessary to support the proposed innovation in E-Health.* | Inquisitiveness of e-oral health technology | Adequate human resources and technical knowledge | *I think that technology is very helpful for us. And I can get any information related to health issues. So, as I said, I have laptop mobile and internet connection. So, I think it is very helpful for me.* |
| | | | *I think personally I will say I have every access, you know, the eating program treated within and travel program readily available to me and I have even the educational system mathematics is scientifically strategy, and everything is for me, I have an access: same way this will be an additional access.* |

saturation level. A total of four themes emerged from the analysis: unlocking barriers, E-Oral Health awareness, inquisitiveness for E-Oral Health technology and enduring oral health benefits. These themes cover all three types of readiness; core, engagement and structural readiness.

## 1. Unlocking barriers

This theme covers core readiness, as participants expressed their needs for E-Oral Health services by expressing dissatisfaction with the current oral health care system.

The need for E-Oral Health services emerged from participants' previous experiences of oral health services and the challenges that they faced in accessing dental care. Most of the participants identified multiple barriers, such as being recent immigrants, lack of familiarity with the health care system, lack of information, language barriers, financial challenges, not having dental insurance coverage, long waiting hours to see a dentist in the public setting and lack of transportation.

*"Waiting to see Dentist is always been concern in Canada, ever since I am here, I faced so many problems like speaking French"* (Participant 6, Interview).

*"I'm really running short of the information where to go, where not to go to collect the data, where what and when it should be done"* (Participant 11, Interview).

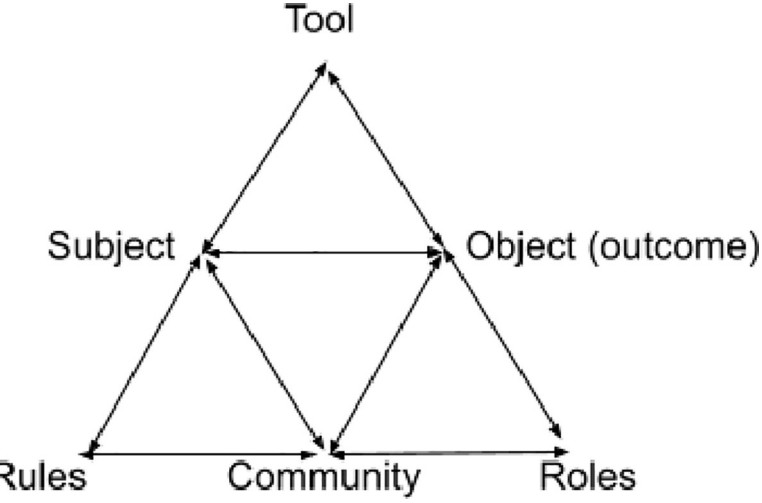

**Fig 1. Activity theory.**

**Table 2. Elements of Activity theory adapted to e-oral health technology.**

| Element | An example of the element |
|---|---|
| **Subject** | Study Participants |
| **Object** | Explore patient's readiness |
| **Outcome** | E-Readiness |
| **Tools** | E-Oral Health technology |
| **Rules** | Change in Environment, such as immigrants moving to a new country, its culture and health system |
| **Community** | Immigrants and Canadians |
| **Roles** | Complexity of access to care |

Participants expressed that E-Oral Health technology may be a potential solution to some of these barriers. They described E-Oral Health technology as a facilitator to improving oral health care and access to care.

*"You know, if the dental services are available for all the time like 24\* 7, Everybody would love to have those services because the health is the kind of thing, things happen. You don't know the time"* (Participant 9, Interview).

*"I think it is innovative idea which could be the facilitator, I think it would definitely improve the oral health care and oral health access to care to the people overall"* (Participant 8, Interview).

## 2. E-Oral health awareness

This theme covers engagement and patient readiness as participants were exposed to the concept of E-Health. Participants actively debated the perceived benefits of E-Oral Health, as well as its disadvantages. They consider its benefits as immediate, providing easy access to information and health care services, as well as being affordable. Participants also deem E-Oral Health

**Table 3. Participants' demographic characteristics.**

| Characteristics | Number of Participants |
|---|---|
| Age | |
| • 20–40 years | 11 |
| • 40–60 years | 2 |
| • 60–80 years | 2 |
| Gender | |
| • Male | 7 |
| • Female | 8 |
| Residential status | |
| • Immigrant | 10 |
| • Born in Canada | 5 |
| Highest level of education attained | |
| • Elementary | 2 |
| • Secondary | 1 |
| • Higher/University | 12 |
| Domestic Status | |
| • Living alone | 8 |
| • Living with partner or child | 7 |

to be cost-effective in oral health care service provision, even at organizational, governmental and policy maker level.

> *"Well, if it will happen, I would be very satisfied. I believe it's a very good future application to be done for people even for citizens or for the newcomers to Canada, it's really helpful"* (Participant 4, Interview).

> *"In Canada, people use banking related applications and to get information like Metro bus service, they use application. So, I think so they will definitely use this kind of application for their health issues"* (Participant 14, Interview).

> *"Yes, I think is 100% affordable because if you do not have at home, you have in the library, you have on joints like Tim Horton and other eating places"* (Participant 3, Interview).

> *"It's (E-Oral Health) easier access. So, it would be easier, it can make your life easier. It could maybe make your life more convenient"* (Participant 5, Interview).

On the contrary, lack of physical interaction with the dentist, technical issues and data privacy issues were expressed as potential disadvantages of E-Oral Health care.

> *"It's even hard for,. . . because you cannot feel that texture, you can't feel the edges. Cameras never going to be good enough for you to see it. Even if you're increasing . . . lighting and special magnifying glasses. Oral Health is really hard to show to inside of your mouth through a camera"* (Participant 12, Interview).

Interestingly, most participants were aware of E-Oral Health and considered it to be an interesting technology.

> *"To be honest with you, I haven't heard about that before, but it seems like a good idea"* (Participant 1, Interview).

### 3. Inquisitiveness to learn and use E-Oral health technology

This theme covers engagement and structural readiness, as participants shared their views of learning E-Oral Health technology and its perceived advantages. Participants were optimistic about obtaining E-Oral Health applications and were ready to pay for such applications because they believe that it would be cost-effective. They thought that this technology is the future of oral health care and expressed their interest, primarily in active learning.

> *"If something like that is there which is specifically prepared for the e-dentistry, I would be happy to learn about that"* (Participant 8, Interview).

> *"As a mother if you ask any mother to this (E-Oral Health training) she will say yes, because that's the kind of very handy, so I can easily access, I can talk to maybe I can text them that this is a problem and what should be the next step"* (Participant 5, Interview).

> *"I mean most app-like ranges and for Apple there $1 each or whatever, $2, even if it goes up to $10, as long as it does the job, people will pay for it"* (Participant 7, Interview).

### 4. Enduring oral health benefits

This theme covers core readiness, as participants considered E-Health to be a solution to reducing health care challenges and expressed their beliefs in its long-term benefits.

Participants anticipated that this technology would be promising in reducing oral health inequalities, especially for vulnerable populations including immigrants, refugees and those living in rural and remote areas. They considered it as a technology that can potentially improve oral health literacy and users' satisfaction at both individual and wider societal level.

> *"I think to have more access to the E appraisal of healthcare or Cybernet will be really it will be too good stead for benefits to the society"* (Participant 2, Interview).

> *"It would help everyone in rural remote all the people living in any areas"* (Participant 13, Interview).

## Discussion

A better understanding of e-health is of public health importance since it could lead to the implementation of effective policies based on patients' perceptions and needs [27, 28]. Various E-Health Readiness frameworks have been developed to understand readiness from different stakeholders' perspectives, especially those of health care providers and health organizations [13, 28–31]. Only one among those frameworks included the patients' perspective on E-Health Readiness [13]. Moreover, most of those frameworks lack credible evaluation and validation [28]. To our knowledge, this study is the first to explore the patient-perspective on E-Readiness in the field of oral health. Study results indicate that participants demonstrated their core, engagement and structural readiness for adoption and implementation of E-Oral Health technology within the Canadian health system. They considered this technology effective, not only for themselves and their families, but also for the society at large; however, they also revealed a few barriers that might need to be considered.

Various concepts have been used to elucidate E-Health technology and its readiness, such as Theory of Change and Innovation Diffusion Theory [32]. Among these, the use of Activity Theory in our study was influenced by a previous study that suggested using Activity Theory as a framework for E-Health Readiness assessment in health care institutions [32]. Activity Theory is popular not only in health research, but also in various fields, including information system, education, culture, psychology, management and human technology interaction research [26, 33]. The available literature suggests that Activity Theory is pertinent in cases of understanding and solving problems related to e-readiness and e-learning and their associated environments [26, 32]. Moreover, this theory is coherent with qualitative research methodology due to its holistic and conceptual nature of exploring human activities, such as E-Oral Health technology in this study [26].

Based on our data and elements of activity theory, the activity system of this research work is illustrated by Fig 2. Activity when using E-Oral Health technology to report the result on

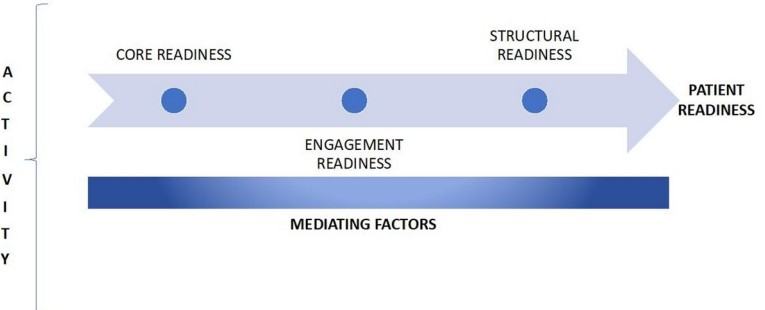

**Fig 2. Activity when using E-Oral health technology to report the result.**

E-Readiness. The Activity Theory allowed us to understand the patients' E-Oral Health Readiness by exploring ongoing activities in different types of readiness at every stage of the study. As per the elements of Activity Theory, the results of this study suggest that E-Oral Health technology, being a central activity tool, prompted dental patients to be ready to use this technology. Their readiness was influenced by various mediating factors, such as their dissatisfaction with the oral health care system, awareness of E-Oral Health and motivation to use this technology.

Patient participation is imperative even earlier in order to effectively design, implement and utilize E-Health technology. A deep understanding of patient needs regarding the use of E-Health and E-Oral Health will aid in these efforts [34]. Patient perspectives on E-Oral Health have been measured among a wide range of patients utilizing the health services in both developed and developing nations, such as in general private and public health services, primary health care services, rehabilitation services and services for multi-morbid chronic diseases [3, 34–38]. Our results are in line with available evidence on patient perspective for E-Health technology in other health disciplines relative to its positive impact on access, treatment adherence, cost-effectiveness, health outcomes, satisfaction, empowerment and quality of life [34, 37–42]. Moreover, these studies on e-health also reported patients' willingness to use and learn such technology, also similar to this present study [3, 34–36, 40, 41]. Furthermore, patients' concerns regarding E-Oral Health were also consistent with that of E-Health technology in terms of lack of human contact and personal data privacy [34, 43].

The results of this study will create a platform in dentistry to develop and validate E-Oral Health readiness instruments for future oral health research. Various recommendations are suggested to optimize the use of technology in oral healthcare practices. For example, the development of E-oral Health technology training programs for its users as well as the creation of E-Oral based applications such as oral health education-based application for children and adults, oral health care access related applications, oral health digital service management, E-consultations. Dentist should recommend such technology to their patients in order to facilitate its use. Simultaneously, detailed policies and legislations should be developed to protect patients' privacy, access and sharing of E-Oral Health related data.

The results of this study can be generalized only to similar settings; further research is necessary to determine whether the results identified in this study are relevant to other populations. Another possible limitation was conducting the interviews only in the English language in Montreal, which is primarily a French-speaking city. This criterion excluded the perceptions of Francophone people. Similarly, another language-based limitation was the inclusion of non-native English speakers who may have had difficulty in expressing their views in the English language. Lastly, the lack of prior awareness of E-Oral Health among the participants suggests the need to introduce and create more E-Oral Health awareness in the public education system. This study prepares the ground for future studies aimed to understand multi-stakeholders' perspectives on E-Oral Health in both developed and developing nations.

## Conclusion

The study results suggest that dental patients consider E-Oral Health to be a facilitator to access to care, and they were ready to learn and use E-Oral Health technology. Implementation of and support for E-Oral Health technologies are needed to improve access to care for many populations.

## Supporting information

**S1 File. Interview guide.**
(DOCX)

## Acknowledgments

Authors would like to express their sincere and profound gratitude to all the participants, and research team members for their constant help and support.

## Author Contributions

**Conceptualization:** Arishdeep Kaur Jagde, Jocelyne Feine, Elham Emami.

**Data curation:** Arishdeep Kaur Jagde, Elham Emami.

**Formal analysis:** Arishdeep Kaur Jagde, Richa Shrivastava, Elham Emami.

**Methodology:** Arishdeep Kaur Jagde, Richa Shrivastava, Jocelyne Feine, Elham Emami.

**Project administration:** Jocelyne Feine, Elham Emami.

**Resources:** Arishdeep Kaur Jagde, Richa Shrivastava, Elham Emami.

**Supervision:** Jocelyne Feine, Elham Emami.

**Validation:** Arishdeep Kaur Jagde, Richa Shrivastava, Jocelyne Feine, Elham Emami.

**Visualization:** Richa Shrivastava, Jocelyne Feine, Elham Emami.

**Writing – original draft:** Arishdeep Kaur Jagde, Elham Emami.

**Writing – review & editing:** Arishdeep Kaur Jagde, Richa Shrivastava, Jocelyne Feine, Elham Emami.

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
