## [Decision Letter · Decision Letter 0]

4 May 2021

PONE-D-21-09215

Patients’ e-readiness to use e-health technologies for oral health

PLOS ONE

Dear Dr. Emami,

Thank you for submitting your manuscript to PLOS ONE. After careful consideration, we feel that it has merit but does not fully meet PLOS ONE’s publication criteria as it currently stands. Therefore, we invite you to submit a revised version of the manuscript that addresses the points raised during the review process.

We look forward to receiving your revised manuscript.

Kind regards,

Frédéric Denis, Ph.D.

Academic Editor

PLOS ONE

Journal Requirements:

"Authors would like to express their sincere and profound gratitude to all the participants, and

research team members for their constant help and support. Dr. Elham Emami is supported by

a Clinician Scientist Award from the Canadian Institute of Health Research."

"NO - The funders had no role in study design, data collection and analysis, decision to publish, or preparation of the manuscript."

Reviewers' comments:

Reviewer's Responses to Questions

**Comments to the Author**

1. Is the manuscript technically sound, and do the data support the conclusions?

Reviewer #1: Yes

Reviewer #2: Yes

2. Has the statistical analysis been performed appropriately and rigorously? 

Reviewer #1: N/A

Reviewer #2: Yes

3. Have the authors made all data underlying the findings in their manuscript fully available?

Reviewer #1: Yes

Reviewer #2: Yes

4. Is the manuscript presented in an intelligible fashion and written in standard English?

Reviewer #1: Yes

Reviewer #2: No

5. Review Comments to the Author

Reviewer #1: Dear author,

Thank you for your work and your article. I think that this study is very intersting. I know that the evaluation of patients' e-readiness is an important point to implement digital oral health. The methodology of this kind of study is always difficult but the one you choose seems adapted. The most important part for implementation of digital health is the organizationnal aspect and it is always forgotten. Your study works on the point of view of patients and users and it is the beginning of this kind of programme.

Because you only received 15 interviews of course this study could not be used to generalized the idea but it has to be done as the first step.

I found some typo mistakes

Reviewer #2: Thank you for giving me a chance to read such an interesting manuscript, It has a uniqueness in itself but I would like to clear certain doubts and Authors should thoroughly revise the manuscript for better understanding and readability.

Overall in whole manuscript, Grammatical mistakes can be observed. Please revise the manuscript with standard English and give full consideration for Grammar and Syntax while writing. Authors should avoid use of long sentences and symbols in sentence. Use of too many "and" in a sentence should also be avoided. This will enhance the readability of the manuscript.

Please go through the following points:

Abstract

“Four major themes emerged from the study: Unlocking barriers, E-oral Health Awareness, Inquisitiveness for e-oral health technology and enduring oral health Benefits” Authors should avoid using “:” colon symbol while writing. It should be in sentence form.

Authors wrote “A total of 15 face-to-face, semi-structured, 60 to 90-minute audio-recorded interviews

were conducted.” Incorrect

“A total of 15 face-to-face, semi-structured and 60 to 90-minute audio recorded interviews were conducted.” Correct

Keywords should be in accordance to MeSH terms. Authors wrote E-Readiness, E-health, E-Oral Health in keywords whereas in full manuscript somewhere it is written capital “E” and somewhere small “e”. It should be uniform in whole manuscript.

Introduction

Page no 4, “Jennett et al. have introduced three e-readiness domains: (1) Core readiness refers……” Authors should write either is paragraph format or point wise format. Just an example to write in paragraph format “Jennett et al. have introduced three e-readiness domains: core readiness, engagement readiness and structural readiness.” After this line explain each domain individually.

Rationale is very well explained but I feel author should avoid using such statement in abstract and manuscript “To our knowledge, there is no study that specifically examines e-readiness in the discipline of dentistry from a patient’s perspective.”

Materials and methods

Ethical approval statement and participant consent for the study should be mentioned in start of materials and methods section and not in data analysis section. Authors should also provide ethical approval registration number.

Authors should mention Inclusion and exclusion criteria for participants

Instead of term snowball technique authors should use more universally acceptable terminology that is “snowball sampling or snowball sampling technique”. Rationale for snowball sampling? Please mention the same in discussion. As the population for the current study was easily accessible so why snowball sampling was used.

“Data were coded manually, then analyzed using ATLAS.ti to facilitate the analysis.” Authors should reframe this sentence and they should be specific while writing “ATLAS.ti” because way of writing “ATLAS.ti” is different which is been mentioned in abstract. There should be uniformity while writing it. If it is a software proper citation along with version should be mentioned.

Page no 6, “I expect high standard of high quality of all health care and I think I would say with the most modern life technology, in terms in terms of the quality of healthcare will improve.” in terms is repeated twice

Page no 7, line no 3 and 4, “it should be cost effective” is repeated twice.

For citation of figure and table, figure and table legend is not needed in main text, just cite it as Figure 1 or Table 1. Please follow this in full manuscript

Results

Very well written.

Page no 11 last line “security/privacy issues” write in sentence form, avoid using slash symbol

Discussion

On page no 15, authors wrote about recommendation for e-health, it should be written in paragraph format. Point wise writing is not appreciable in discussion of manuscript.

References

Authors should strictly follow journals referencing style. Please go through authors instruction for referencing style. It lacks Orthography. Even DOI number is missing for all references

6. PLOS authors have the option to publish the peer review history of their article (what does this mean?). If published, this will include your full peer review and any attached files.

Reviewer #1: No

Reviewer #2: No

---

## [Author Response · Author response to Decision Letter 0]

4 Jun 2021

RESPONSE TO REVIEWERS’ COMMENTS:

REVIEWER 1:

Comment: Thank you for your work and your article. I think that this study is very interesting. I know that the evaluation of patients' e-readiness is an important point to implement digital oral health. The methodology of this kind of study is always difficult but the one you choose seems adapted. The most important part for implementation of digital health is the organizational aspect and it is always forgotten. Your study works on the point of view of patients and users and it is the beginning of this kind of programme. Because you only received 15 interviews of course this study could not be used to generalized the idea but it has to be done as the first step. I found some typo mistakes

Response: We would like to thank the reviewer for the time and constructive comments. We have corrected the typo mistakes.

REVIEWER 2:

Comment: Thank you for giving me a chance to read such an interesting manuscript. It has a uniqueness in itself, but I would like to clear certain doubts and Authors should thoroughly revise the manuscript for better understanding and readability. Overall, in whole manuscript, Grammatical mistakes can be observed. Please revise the manuscript with standard English and give full consideration for Grammar and Syntax while writing. Authors should avoid use of long sentences and symbols in sentence. Use of too many "and" in a sentence should also be avoided. This will enhance the readability of the manuscript.

Response: We would like to thank the reviewer for the time and valuable comments.

Comment 1: Abstract “Four major themes emerged from the study: Unlocking barriers, E-oral Health Awareness, Inquisitiveness for e-oral health technology and enduring oral health Benefits” Authors should avoid using “:” colon symbol while writing. It should be in sentence form.

Response: Thanks for the comment. We have now corrected this issue. Please see page number 2

Comment 2: Authors wrote “A total of 15 face-to-face, semi-structured, 60 to 90-minute audio-recorded interviews were conducted.” Incorrect.

“A total of 15 face-to-face, semi-structured and 60 to 90-minute audio recorded interviews were conducted.” Correct

Response: Thanks for the comment. The suggested change has been done. Please see page number 2.

Comment 3: Keywords should be in accordance to MeSH terms. Authors wrote E-Readiness, E-health, E-Oral Health in keywords whereas in full manuscript somewhere it is written capital “E” and somewhere small “e”. It should be uniform in whole manuscript.

Response: Thanks for the comment. All the ‘e’s are made uniform in the whole manuscript.

Comment 4: Introduction, Page no 4, “Jennett et al. have introduced three e-readiness domains: (1) Core readiness refers……” Authors should write either is paragraph format or point wise format. Just an example to write in paragraph format “Jennett et al. have introduced three e-readiness domains: core readiness, engagement readiness and structural readiness.” After this line explain each domain individually.

Response: Thanks for the comment. On page number 4, we have done the changes as per suggestion. 

Comment 5: Rationale is very well explained but I feel author should avoid using such statement in abstract and manuscript “To our knowledge, there is no study that specifically examines e-readiness in the discipline of dentistry from a patient’s perspective.”

 Response: thanks for the comment. We have done the needful on page number 4.

Comment 6: Materials and methods; Ethical approval statement and participant consent for the study should be mentioned in start of materials and methods section and not in data analysis section. Authors should also provide ethical approval registration number.

Response: Thank you for the comment. As per your suggestion, we have now added the required information to the article on page number 4.

Comment 7: Authors should mention Inclusion and exclusion criteria for participants

Response: Thank you for the comment. As per your suggestion, we have now added the required information to the article on page number 5.

Comment 8: Instead of term snowball technique authors should use more universally acceptable terminology that is “snowball sampling or snowball sampling technique”. Rationale for snowball sampling? Please mention the same in discussion. As the population for the current study was easily accessible so why snowball sampling was used.

Response: Thanks for the comment. As per your suggestion, we have now added the required information to the article on page number 5.

Comment 9: “Data were coded manually, then analyzed using ATLAS.ti to facilitate the analysis.” Authors should reframe this sentence and they should be specific while writing “ATLAS.ti” because way of writing “ATLAS.ti” is different which is been mentioned in abstract. There should be uniformity while writing it. If it is a software proper citation along with version should be mentioned.

Response: Thanks for your comment. We have now made additions to address this issue on page number 6.

Comment 10: Page no 6, “I expect high standard of high quality of all health care and I think I would say with the most modern life technology, in terms in terms of the quality of healthcare will improve.” in terms is repeated twice.

Response: Thanks for the comment. We have now corrected this issue on page number 6.

Comment 11: Page no 7, line no 3 and 4, “it should be cost effective” is repeated twice.

Response: Thanks for the comment. We have now corrected this issue. Please see page number 6.

Comment 12: For citation of figure and table, figure and table legend is not needed in the main text, just cite it as Figure 1 or Table 1. Please follow this in full manuscript.

Response: Thanks for the comment. We addressed this comment in the whole manuscript for this study on page number 8 and 9.

Comment 13: Results Very well written.Page no 11 last line “security/privacy issues” write in sentence form, avoid using slash symbol

Response: Thanks for the comment. The suggested change has been done.

Comment 14: Discussion On page no 15, authors wrote about recommendation for e-health, it should be written in paragraph format. Point wise writing is not appreciable in discussion of manuscript.

Response: Thanks for the comment. We addressed this comment in the discussion section for this study. Please see page number 15 and 16.

Comment 15: References Authors should strictly follow journals referencing style. Please go through authors instruction for referencing style. It lacks Orthography. Even DOI number is missing for all references

Response: Thanks for the comment. We have now corrected this issue.

---

## [Decision Letter · Decision Letter 1]

16 Jun 2021

Patients’ E-Readiness to use E-Health technologies for oral health

PONE-D-21-09215R1

Dear Dr. Emami,

We’re pleased to inform you that your manuscript has been judged scientifically suitable for publication and will be formally accepted for publication once it meets all outstanding technical requirements.

Kind regards,

Frédéric Denis, Ph.D.

Academic Editor

PLOS ONE

Additional Editor Comments (optional):

Reviewers' comments:

Reviewer's Responses to Questions

**Comments to the Author**

1. If the authors have adequately addressed your comments raised in a previous round of review and you feel that this manuscript is now acceptable for publication, you may indicate that here to bypass the “Comments to the Author” section, enter your conflict of interest statement in the “Confidential to Editor” section, and submit your "Accept" recommendation.

Reviewer #2: All comments have been addressed

2. Is the manuscript technically sound, and do the data support the conclusions?

Reviewer #2: Yes

3. Has the statistical analysis been performed appropriately and rigorously? 

Reviewer #2: Yes

4. Have the authors made all data underlying the findings in their manuscript fully available?

Reviewer #2: Yes

5. Is the manuscript presented in an intelligible fashion and written in standard English?

Reviewer #2: Yes

6. Review Comments to the Author

Reviewer #2: I would like to thank authors that they thoroughly revised whole manuscript and considered my suggestions for the same.

7. PLOS authors have the option to publish the peer review history of their article (what does this mean?). If published, this will include your full peer review and any attached files.

Reviewer #2: No

---

## [Editor Report · Acceptance letter]

1 Jul 2021

PONE-D-21-09215R1 

Patients’ E-Readiness to use E-Health technologies for oral health 

Dear Dr. Emami:

I'm pleased to inform you that your manuscript has been deemed suitable for publication in PLOS ONE. Congratulations! Your manuscript is now with our production department. 

Kind regards, 

on behalf of

Dr. Frédéric Denis 

Academic Editor

PLOS ONE